# MULTI-CLASS CLASSIFICATION WITHOUT MULTI-CLASS LABELS

**Yen-Chang Hsu[1], Zhaoyang Lv[1], Joel Schlosser[2], Phillip Odom[2], and Zsolt Kira[12]**
[1]Georgia Institute of Technology
[2]Georgia Tech Research Institute
[1]{yenchang.hsu,zhaoyang.lv,zkira}@gatech.edu
[2]{joel.schlosser,phillip.odom}@gtri.gatech.edu

## ABSTRACT

This work presents a new strategy for multi-class classification that requires no class-specific labels, but instead leverages pairwise similarity between examples, which is a weaker form of annotation. The proposed method, meta classification learning, optimizes a binary classifier for pairwise similarity prediction and through this process learns a multi-class classifier as a submodule. We formulate this approach, present a probabilistic graphical model for it, and derive a surprisingly simple loss function that can be used to learn neural network-based models. We then demonstrate that this same framework generalizes to the supervised, unsupervised cross-task, and semi-supervised settings. Our method is evaluated against state of the art in all three learning paradigms and shows a superior or comparable accuracy, providing evidence that learning multi-class classification without multi-class labels is a viable learning option.

## 1 INTRODUCTION

One of the most common settings for machine learning is classification, which involves learning a function $f$ to map the input data $x$ to a class label $y \in \{1, 2, .., C\}$. The most successful method for learning such a function is deep neural networks, owing its popularity to its capability to approximate a complex nonlinear mapping between high-dimensional data (e.g. images) and the classes. Despite the success of deep learning, a neural network demands a large amount of *class-specific* labels for learning a discriminative model, *i.e.* $P(y|x)$. This type of labeling can be expensive to collect, requires a-priori knowledge of all classes, and limits the form of supervision required. For example, the classes may be ambiguous or non-expert human annotators may be able to more easily provide information about whether two instances are of the same class or not, rather than identifying the specific class. A final problem is that different methods are necessary depending on what type of data is available, ranging from supervised learning (known classes) to cross-task unsupervised learning (unknown classes in the target domain) and semi-supervised learning (mix of labeled and unlabeled with known classes). Unsupervised learning with unknown classes is especially difficult to support.

To relax these limitations, we propose to *reduce* the problem of classification to a meta problem that underlies a set of learning problems. Instead of solving the target task directly (learning a multi-class discriminative model such as a neural network), we instead learn a model that does not require explicit class label $y$ but rather a weaker form of information. In the context of classification, the meta problem that we use is a binary decision problem. Note that such a conversion to a different task (e.g. binary) is called a *problem reduction* method (Allwein et al., 2000) which has had a long history in the literature, especially in ensemble methods and binarization techniques (Galar et al., 2011). The most well-known strategies are "one-vs-all" (Anand et al., 1995; Rifkin & Klautau, 2004) and "one-vs-one" (Knerr et al., 1990; Hastie & Tibshirani, 1998; Wu et al., 2004). Although they have varied ensembling strategies, all of them share the same task encapsulating scheme, as illustrated in Figure 1a; specifically the binary classifiers are the sub-modules of a multi-class classifier (i.e. the multi-class classifier consists of multiple binary classifiers). These schemes still require that the class

---

The demo code is available at https://github.com/GT-RIPL/L2C

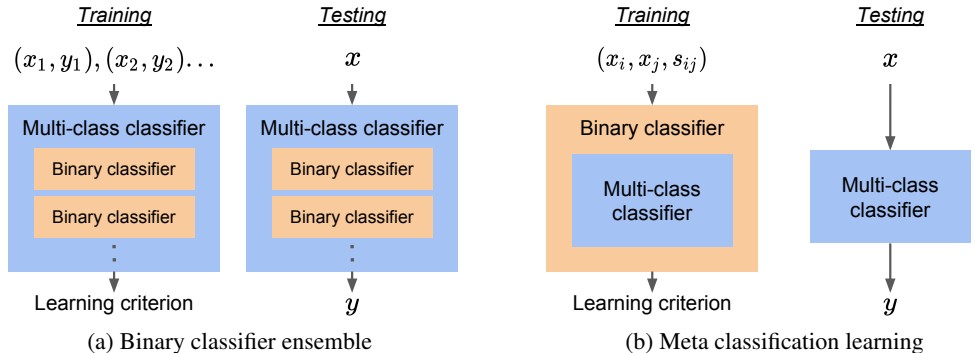

Figure 1: Problem reduction schemes for multi-class classification. This work proposes scheme (b), which introduces a binary classifier that captures $s_{ij}$. Note that $s_{ij}$ represents the probability that $x_i$ and $x_j$ belong to the same class.

label $y$ be available to create the inputs for each binary classifier, and therefore these strategies do not relax the labeling requirements mentioned earlier.

In this work, we propose a novel strategy to address the above limitations. Our method reverses the task encapsulation order so that a multi-class classifier becomes a sub-module of a binary classifier, as illustrated in Figure 1b. The connection between the two classifiers is elucidated in Section 3. There are two highlights in Figure 1b. First, class labels $y_i$ are not required in the learning stage. Instead, our method uses pairs of data $(x_i, x_j)$ as input and pairwise similarity $s_{ij}$ for the supervision. Second, there is only one binary classifier in the scheme and it is present only during the training stage. In other words, the ephemeral binary task assists the learning of a multi-class classifier without being involved in the inference. When using a neural network with softmax outputs for the multi-class classifier, the proposed scheme can learn a discriminative model with only pairwise information.

We specifically make the following contributions: 1) We analyze the problem setting and show that the loss we can use for this encapsulation can be easily derived, and we present an intuitive probabilistic graphical model interpretation for doing so, 2) We evaluate its performance compared to vanilla supervised learning of neural networks which uses multi-class labels, and visualize the loss landscape to better understand the underlying optimization difficulty, and 3) We demonstrate support for learning classifiers in more challenging problem domains, e.g. in unsupervised cross-task transfer and semi-supervised learning. We show how our meta classification framework can support all three learning paradigms, and evaluate it against several state-of-the-art methods. The experimental results show that the same meta classification approach is superior or comparable to state of the art across the three problem domains (supervised learning, unsupervised cross-task learning, and semi-supervised learning), demonstrating flexibility to support even unknown types and numbers of classes.

## 2 RELATED WORK

**Supervised learning and problem reduction:** Allwein et al. (2000) presents a unifying framework for multi-class classification by reducing it to multiple binary problems. The concepts for achieving such reduction, one-vs-all and one-vs-one, have been widely adopted and analyzed (Galar et al., 2011). The two strategies have been used to create several popular algorithms, such as variants of support vector machine (Weston & Watkins, 1998), AdaBoost (Freund & Schapire, 1997; Schapire & Singer, 1999), and decision trees (Fürnkranz, 2003). Despite the long history of reduction, our proposed scheme (Figure 1b) has not been explored. Furthermore, the scheme can be deployed easily by replacing the learning objective, which is fully compatible with deep neural networks for classification, a desirable property for broad applicability.

**Unsupervised cross-task transfer learning:** This learning scheme is proposed by Hsu et al. (2018). The method transfers the pairwise similarity as the meta knowledge to an unlabeled dataset of different classes. It then uses a constrained clustering algorithm with predicted pairwise constraints

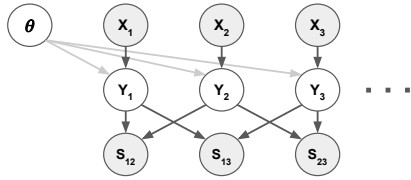

Figure 2: Graphical representation for the meta classification task; $X_i$ represents the node of input data, $Y_i$ represents the class label, $S_{ij}$ is pairwise similarity between instances $i$ and $j$, and $\theta$ represents the neural network parameters.

(binarized pairwise similarity) to discover the unseen classes. This learning scheme shares the same supervision (pairwise similarity) as ours, and therefore relates our method to constrained clustering algorithms. One class of such approaches uses the constraints to learn a distance metric, and then applies a generic clustering algorithm such as K-means or hierarchical clustering to obtain the cluster assignments. This includes DML (Xing et al., 2003), ITML (Davis et al., 2007), SKMS (Anand et al., 2014), SKKm (Anand et al., 2014; Amid et al., 2016), and SKLR (Amid et al., 2016). The second class of methods incorporates the constraints into the cluster assignment objective. Some constrained spectral clustering algorithms, *e.g.* CSP (Wang et al., 2014) and COSC (Rangapuram & Hein, 2012) use this strategy. There are also approaches combine both distance metric learning and a clustering objective jointly, such as MPCKMeans (Bilenko et al., 2004), CECM (Antoine et al., 2012), and Kullback–Leibler divergence based contrastive loss (KCL) (Hsu & Kira, 2016; Hsu et al., 2018). Our new learning objective for Figure 1b can replace the above objectives in the cross-task transfer learning scheme.

**Semi-supervised learning:** Our meta classification strategy can easily plug into a semi-supervised learning scheme. Our comparison focuses on state-of-the-art methods which solely involve adding a consistency regularization (Laine & Aila, 2017; Sajjadi et al., 2016; Miyato et al., 2018; Tarvainen & Valpola, 2017) or Pseudo-Labeling (Lee, 2013) for training a neural network. Another line of strategy combines weak supervision, such as similar pairs, and unlabeled data to learn a binary classifier (Bao et al., 2018). We present a new method by augmenting Figure 1b with the Pseudo-Labeling strategy.

## 3 META CLASSIFICATION LEARNING

A natural way to analyze problems with observed and unobserved information is through a probabilistic graphical model. In figure 2, we show the graphical model for our problem, where class-specific labels $Y$ are latent while pairwise similarities $S$ are observed. Specifically, we denote $\mathbf{X} = \{X_1, .., X_n\}$, $\mathbf{Y} = \{Y_1, .., Y_n\}$, and $\mathbf{S} = \{S_{ij}\}_{1 \le i, j \le n}$ to represent the nodes for samples, class labels, and pairwise similarities, respectively. In the model, we have $Y_i \in \{1, 2, .., C\}$ and $S_{ij} \in \{0, 1\}$. Then we have $\mathrm{P}(S_{ij} = 1 | Y_i, Y_j) = 1$ when $Y_i = Y_j$ and zero probability otherwise; similarly, $\mathrm{P}(S_{ij} = 0 | Y_i, Y_j) = 1$ when $Y_i \ne Y_j$. The output of a discriminative classifier with parameters $\theta$ is $f(x_i; \theta) = \mathrm{P}(Y_i | x_i; \theta)$, where $f(x_i; \theta)$ outputs a categorical distribution. Now we describe the likelihood that the model explains the observed labeling (either with class labeling or pairwise labeling).

$$\mathcal{L}(\theta; \mathbf{X}, \mathbf{Y}, \mathbf{S}) = \mathrm{P}(\mathbf{X}, \mathbf{Y}, \mathbf{S}; \theta) = \mathrm{P}(\mathbf{S}|\mathbf{Y})\mathrm{P}(\mathbf{Y}|\mathbf{X}; \theta)\mathrm{P}(\mathbf{X}) \tag{1}$$

When $\mathbf{S}$ is fully observed while $\mathbf{Y}$ is unknown, calculating the likelihood requires marginalizing $\mathbf{Y}$ by computing $\sum_{\mathbf{Y}} \mathrm{P}(\mathbf{S}|\mathbf{Y})\mathrm{P}(\mathbf{Y}|\mathbf{X}; \theta)$, which is intractable. The pairwise term $\mathrm{P}(\mathbf{S}|\mathbf{Y}) = \prod_{i,j} \mathrm{P}(S_{ij}|Y_i, Y_j)$ makes all $Y_i$ dependent on each other and prohibits efficient factorization. Thus, we approximate the computation by imposing additional independences such that $S_{ij} \perp \mathbf{S} \setminus \{S_{ij}\} | X_i, X_j$ (see Appendix D for a discussion of these). Now we can compute the likelihood with the observed nodes $X_i = x_i$ and $S_{ij} = s_{ij}$:

$$\mathcal{L}(\theta; \mathbf{X}, \mathbf{S}) \approx \sum_{\mathbf{Y}} \mathrm{P}(\mathbf{S}|\mathbf{Y})\mathrm{P}(\mathbf{Y}|\mathbf{X}; \theta) \tag{2}$$

$$\approx \prod_{i,j} \Big( \sum_{Y_i = Y_j} \mathbb{1}[s_{ij} = 1]\mathrm{P}(Y_i|x_i; \theta)\mathrm{P}(Y_j|x_j; \theta) +$$

$$\sum_{Y_i \neq Y_j} \mathbb{1}[s_{ij} = 0]\mathrm{P}(Y_i|x_i; \theta)\mathrm{P}(Y_j|x_j; \theta) \Big). \tag{3}$$

Equation (2) omits the $\mathrm{P}(\mathbf{X})$ since $\mathbf{X}$ are observed leaf nodes. It is straightforward to take a negative logarithm on equation 3 and derive a loss function:

$$L_{meta}(\theta) = -\sum_{i,j} \log \Big( \sum_{Y_i = Y_j} \mathbb{1}[s_{ij} = 1]\mathrm{P}(Y_i|x_i; \theta)\mathrm{P}(Y_j|x_j; \theta) +$$

$$\sum_{Y_i \neq Y_j} \mathbb{1}[s_{ij} = 0]\mathrm{P}(Y_i|x_i; \theta)\mathrm{P}(Y_j|x_j; \theta) \Big) \tag{4}$$

$$= -\sum_{i,j} s_{ij} \log(f(x_i; \theta)^T f(x_j; \theta)) + (1 - s_{ij}) \log(1 - f(x_i; \theta)^T f(x_j; \theta)). \tag{5}$$

Then we define the function $g$ by the probability of having the same class label, which is calculated by the inner product between two categorical distributions:

$$g(x_i, x_j, f(\cdot, \theta)) = f(x_i; \theta)^T f(x_j; \theta) = \hat{s}_{ij} \tag{6}$$

Here we use $\hat{s}_{ij}$ to denote the predicted similarity (as opposed to ground truth similarity $s_{ij}$). By plugging equation 6 into equation 5, $L_{meta}$ has the form of a binary cross-entropy loss:

$$L_{meta} = -\sum_{i,j} s_{ij} \log \hat{s}_{ij} + (1 - s_{ij}) \log(1 - \hat{s}_{ij}). \tag{7}$$

In Figure 1b, the multi-class classifier corresponds to $f$ while the binary classifier corresponds to $g$. In other words, it is surprisingly simple to wrap a multi-class classifier by a binary classifier as described above. Since there are no learnable parameters in $g$, the weights optimized with the meta criterion $L_{meta}$ are all in the neural network $f$. To minimize $L_{meta}$, $f(x_i; \theta)$ and $f(x_j; \theta)$ must output a sharply peaked distribution with the peak happening only at the same output node when $s_{ij} = 1$. In the case of $s_{ij} = 0$, the two distributions must have as little overlap as possible to minimize the loss. In the latter case, the two samples are pushed to be activated at the output nodes of different classes. Both properties of $f$'s output distribution are typical characteristics of a classifier learned with class labels and using multi-class cross-entropy. The properties also illustrate the intuition of why minimizing $L_{meta}$ helps $f$ learn outputs similar to a multi-class classifier.

Lastly, because of the likelihood nature of $L_{meta}$, we call the learning criterion a Meta Classification Likelihood (MCL) in the rest of the paper.

## 4 LEARNING PARADIGMS

The supervision used in MCL is the pairwise labeling $S$. Due to its weaker form compared to class labels, we have the flexibility to collect it in a supervised, cross-task transfer, or semi-supervised manner. The collection method determines the learning paradigms. In the first two learning paradigms, other methods (see Related Work Section) have also used pair-wise constraints similarly; our novelty is the derivation of our new learning objective, MCL, which can replace the other objectives. In the semi-supervised learning scenario, the proposed Pseudo-MCL is a new method. Details are elaborated below.

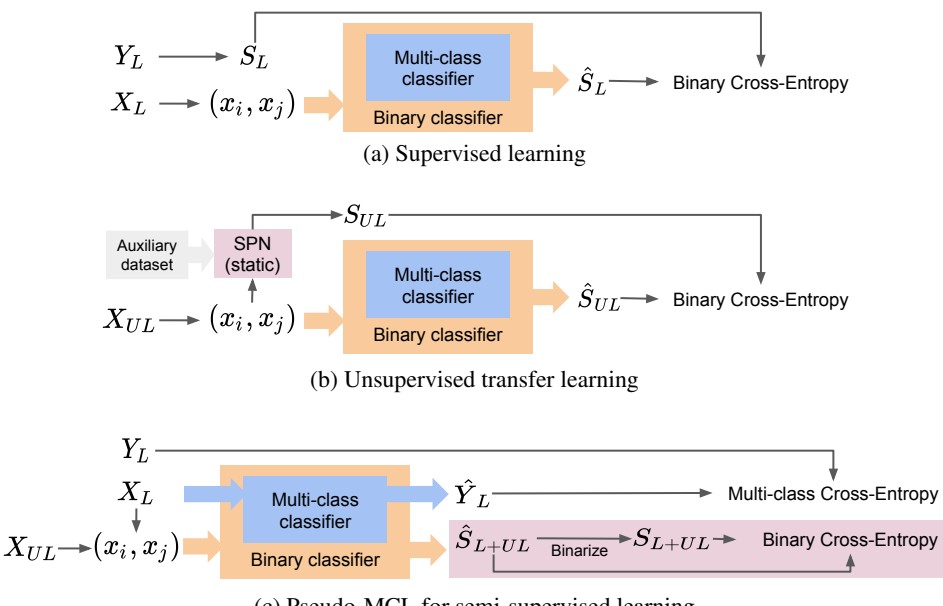

Figure 3: The training flows for each learning paradigm. $X_L$ represents the labeled data with class label $Y_L$. $X_{UL}$ is unlabeled data. $\hat{S}$ is the predicted pairwise similarity while $S$ is used as the learning target. The similarity prediction network (SPN) in (b) is learned on a labeled auxiliary dataset and transferred to the target dataset $X_{UL}$.

## 4.1 SUPERVISED LEARNING

Supervised pairwise labeling can be directly collected from humans, or converted from existing class labeling by having $S = \{s_{ij}\}_{1 \leq i,j \leq n}$, where $s_{ij} = 1$ if $x_i$ and $x_j$ belong to the same class, otherwise $s_{ij} = 0$. In our experiments, we use the latter setting to enable comparison to other supervised algorithms. Figure 3a illustrates the training process.

## 4.2 UNSUPERVISED LEARNING

Pairwise labeling can come from several natural cues, such as spatial and temporal proximity. For example, the patches in an image can be similar because of their spatial closeness, and the frames of video in a short time usually have similar content. Additionally, useful pairwise information can be found in the edges in social networks or in the network of academic citations. All of the above are potential applications of this work.

Another strategy that is unsupervised in the target domain is to collect pairwise labels through transfer learning. Hsu et al. (2018) proposes a method in which a similarity prediction network (SPN) can be learned from a labeled auxiliary dataset. Then the SPN is applied on the unlabeled target dataset to predict $S$ (the probability of being in the same class). In the last step, the predicted $S$ is fed into a network (in that case optimized via Kullback–Leibler divergence based contrastive loss) to discover the categories in the unlabeled target dataset. Figure 3b illustrates above process. Note that the classes between the auxiliary dataset and target dataset may have an overlap (cross-domain transfer) or not (cross-task transfer) (Hsu et al., 2018). In both cases, the predicted pairwise similarity is noisy (especially in the latter case); therefore the transfer learning strategy creates a challenging scenario for learning classifiers. Its difficulty makes it a good benchmark to evaluate the robustness of our methods and is used in our experiments.

## 4.3 SEMI-SUPERVISED LEARNING

We propose a new strategy to obtain the $S$ for semi-supervised learning. Figure 3c illustrates the method under the typical semi-supervised learning setting, which takes a common dataset $D$ used for

supervised learning and discards the labels for most of the dataset. The labeled and unlabeled portions in $D$ are $D_L = (X_L, Y_L)$ and $D_{UL} = X_{UL}$ correspondingly. The main idea is to create a pseudo-similarity $S_{L+UL}$ for the meta classifier (similar to Pseudo-Labeling (Lee, 2013)) by binarizing the predicted $\hat{S}_{L+UL}$ at probability 0.5. We call the method Pseudo-MCL, and we note that here interestingly $g$ is not static as it iteratively improves as $f$ improves. Another way to create similarity is data augmentation, inspired by the Π-model (Laine & Aila, 2017) or Stochastic Perturbations (Sajjadi et al., 2016). An image perturbed in different ways naturally belong to the same class, and thus provides free ground-truth similarity. The similarity from both methods can be easily combined to $S_{L+UL}$ by having a logical-OR operation for the two binarized similarities. The learning objective is the sum of the multi-class cross-entropy and Pseudo-MCL, so the mapping between output nodes and classes are automatically decided by the supervised part of learning.

## 5 Experiments

### 5.1 Experimental Setup and Network Optimization

In all experiments, we use a standard gradient-based method for training a neural network by optimizing the learning criterion. For example, with stochastic gradient descent, we calculate MCL within a mini-batch of data. In that case, the $i$ and $j$ correspond to the index of data in a mini-batch $b$. The outputs of $f(\cdot; \theta)$ are enumerated in $|b|(|b| - 1)/2$ pairs in a mini-batch before calculation of MCL. Our empirical finding is that this enumeration introduces a negligible overhead to the training time. We also note that for large datasets, this only samples from the full set of pairwise information.

One limitation of learning a classifier without class labels is losing the mapping (the identifiability) between the output nodes and the semantic class. A simple method to obtain the mapping is by using a part of the training data with class labels and assigning the output nodes to the dominant class which activates the node (here we obtain the optimal assignment by the Hungarian algorithm (Kuhn, 1955), which is commonly used in evaluating the clustering accuracy (Yang et al., 2010)). Note, however, that for unsupervised problems we do not need to do this except to quantitatively evaluate our method; otherwise the outputs can be seen as arbitrary clusters.

### 5.2 Supervised Learning with Weak Labels

This section empirically compares MCL to multi-class cross-entropy (CE) and the strong baseline using pairwise similarity (Kullback–Leibler divergence based contrastive loss (KCL) (Hsu & Kira, 2016; Hsu et al., 2018)), in a supervised learning setting. Specifically, we would like to demonstrate that we can achieve similar classification rates as cross-entropy (the standard objective for multi-class classification) using only pairwise similarity, and show that the previous pairwise criterion cannot do this likely due to a poor loss landscape. We compare the classification accuracy of these criteria with varied network depths and varied dataset difficulty. The visualization of loss landscape is provided in Appendix A. The formulation of KCL and how it relates to MCL is available in Appendix B.

#### 5.2.1 Quantitative analysis

We compare the classification accuracy on three image datasets: MNIST (LeCun, 1998) is a 10-class handwritten digit dataset with 60000 images for training, and 10000 for testing; CIFAR10 and CIFAR100 (Krizhevsky, 2009) instances are colored $32 \times 32$ images of objects such as cat, dog, and ship. They both have 50000 images for training and 10000 for testing.

**Network Architectures:** We use convolution neural networks with a varied number of layers: LeNet (LeCun et al., 1998) and VGG (Simonyan & Zisserman, 2014). We add VGG8, which only has one convolution layer before each pooling layer, as the supplement between LeNet and VGG11. The list of architectures also includes ResNet (He et al., 2016a) with pre-activation (He et al., 2016b)). The number of output nodes $K$ in the last fully connected layer is set to the true number of categories for this section. Since the learning objectives KCL and MCL both work on pairs of inputs, we have a pairwise enumeration layer (Hsu et al., 2018) between the network outputs and the loss function.

**Training Configurations:** All networks in this section are trained from scratch with randomly initialized weights. By default, we use Adam (Kingma & Ba, 2014) to optimize the three criterion

Table 1: The classification error rate (lower is better) on three datasets with different objective functions and different neural network architectures. CE denotes that the network uses class-specific labels for training with a multi-class cross-entropy. MCL only uses the binarized similarity for learning with the meta-classification criterion. KCL is a strong baseline which also uses binarized similarity. The * symbol indicates the worst cases of KCL. The performance in parenthesis means its network uses a better initialization (VGG16 and VGG8) or a learning schedule which is 10 times longer (VGG11). The two treatments are discussed in Section 5.2.1. We only use VGG8 for CIFAR100 since KCL performs the best with it on CIFAR10. Each value is the average of 3 runs.

| Dataset | #class | Network | (Class label) CE | (Pairwise label) KCL | MCL |
|---|---|---|---|---|---|
| MNIST | 10 | LeNet | 0.6% | **0.5%** | 0.6% |
| CIFAR10 | 10 | LeNet | 14.9% | 16.4% | 15.1% |
| | | VGG8 | 10.2% | 10.2% | 10.2% |
| | | VGG11 | 8.9% | 72.2(10.4)% | 9.4% |
| | | VGG16 | 7.6% | *81.1(10.3)% | 8.3% |
| | | ResNet18 | 6.7% | 73.8% | 6.6% |
| | | ResNet34 | 6.6% | 79.3% | 6.3% |
| | | ResNet50 | 6.6% | 79.6% | 5.9% |
| | | ResNet101 | 6.5% | 79.9% | **5.6%** |
| CIFAR100 | 100 | VGG8 | **35.4%** | *45.3(40.2)% | 36.1% |

with mini-batch size 100 and initial learning rate 0.001. On MNIST the learning rate was dropped every 10 epochs by a factor of 0.1 with 30 epochs in total. On CIFAR10/100 we use the same setting except that the learning rate is dropped at 80 and 120 epochs with 140 epochs in total. For CIFAR100, the mini-batch size was 1000 and the learning rate dropped at epoch 100 and 150 with 180 epochs in total. In the experiments with ResNet, we use SGD instead of Adam since SGD converges to a higher accuracy when keeping other settings the same as above. The learning rate for SGD starts with 0.1 and decays with a factor of 0.1 at the number of epochs described above.

**Results and discussion:** The results in Table 1 show that MCL achieves similar classification performance as CE with different network depths and three datasets. In contrast, KCL has degenerate performance when the networks are deeper or the dataset is more difficult. This might be due to a limitation of using KL-divergence, specifically that when two probability distributions are the same, the divergence will be zero no matter what the values are. This property may introduce bad local minima or small gradients for learning. To investigate such a perspective, we apply two strategies. First, we use a large learning rate (0.2) with SGD to avoid bad local minima and make the training schedule 10 times longer for exploring the parameter space. This setting helps KCL with VGG11, in that the error rate drops from 72.2% to 10.4%, but not with VGG16 (from 81.1% to 76.8%). In the second strategy, we select the worst conditions (the values with * notion) in Table 1 for KCL and pre-train the networks with only 4k labels with CE to initialize the networks. Then we use KCL with the full training set to finish the training. With a better initialization, KCL can reach a performance close to CE and MCL. The performance is shown with parenthesis in Table 1. The results of both strategies indicate that KCL has bad local minima or plateaus in its loss surface (see Section A in Appendix). Unlike KCL, MCL can converge to a performance close to CE with random initialization in all of our experiments. Furthermore, MCL outperforms CE with a deeper network (error rate 5.6% versus 6.5% with ResNet101). Such a result indicates that MCL is less prone to overfitting (in the Table 1, all ResNets achieve a training error less than 0.1%).

## 5.3 UNSUPERVISED CROSS-TASK TRANSFER LEARNING

The second experiment follows the transfer learning scenario proposed by Hsu et al. (2018) and is summarized in Section 4.2. This scenario has two settings. The first is when the number of output nodes $K$ equal to the number of ground truth classes $C$ in a dataset. This setting is the same as a multi-class classification task, except no labels (both class labels or similarity labels) are provided in the target dataset. The second setting is having an unknown $C$, which is closer to a clustering

Table 2: Unsupervised cross-task transfer learning on Omniglot. The performance (higher is better) is averaged across 20 alphabets (datasets), in which each has 20 to 47 letters (classes). The ACC and NMI without brackets have the number of output nodes $K$ equal to the true number of classes in a dataset, while columns with "(K=100)" represent the case where the number of classes is unknown and a fixed $K = 100$ is used.

| Method | ACC | ACC (K=100) | NMI | NMI (K=100) |
|---|---|---|---|---|
| K-means (MacQueen et al., 1967) | 21.7% | 18.9% | 0.353 | 0.464 |
| LPNMF (Cai et al., 2009) | 22.2% | 16.3% | 0.372 | 0.498 |
| LSC (Chen & Cai, 2011) | 23.6% | 18.0% | 0.376 | 0.500 |
| ITML (Davis et al., 2007) | 56.7% | 47.2% | 0.674 | 0.727 |
| SKKm (Anand et al., 2014) | 62.4% | 46.9% | 0.770 | 0.781 |
| SKLR (Amid et al., 2016) | 66.9% | 46.8% | 0.791 | 0.760 |
| CSP (Wang et al., 2014) | 62.5% | 65.4% | 0.812 | 0.812 |
| MPCK-means (Bilenko et al., 2004) | 81.9% | 53.9% | 0.871 | 0.816 |
| KCL (Hsu et al., 2018) | 82.4% | 78.1% | 0.889 | 0.874 |
| MCL (ours) | **83.3%** | **80.2%** | **0.897** | **0.893** |

problem. One strategy to address the unknown $C$ is to set a large $K$, and we rely on the clustering algorithm to use only a necessary number of clusters to describe the dataset while leaving the extra clusters empty.

We use constrained clustering algorithms as the baselines since they can use the pairwise inputs from a similarity prediction network (SPN) (Hsu et al., 2018). In this section, the same set of binarized pairwise similarity prediction is provided to all algorithms for a fair comparison. The metric in this section is still the classification accuracy. The mapping between output nodes and classes is calculated by the Hungarian algorithm, in which each class only matches to one output node. The unmapped output nodes are all subject to the classification error. We also include the normalized mutual information (NMI) (Strehl & Ghosh, 2002) metric. We use two datasets in the evaluation.

**Omniglot** (Lake et al., 2015): This dataset has 20 images for each of 1623 different handwritten characters. The characters are from 50 different alphabets and were separated into 30 background sets ($Omniglot_{bg}$) and 20 evaluation sets ($Omniglot_{eval}$) by the dataset author. The procedure uses the $Omniglot_{bg}$ set (964 characters in total) to learn the similarity function and applies it to the cross-task transfer learning on the 20 evaluation sets (this same input is used for all compared algorithms). In this test, the backbone network for classification has four convolution layers and has weights randomly initialized. Both MCL and KCL are optimized by Adam with mini-batch size 100.

**ImageNet** (Deng et al., 2009): The 1000-class dataset is separated into 882-class and 118-class subsets as the random split in Vinyals et al. (2016). The procedure uses $ImageNet_{882}$ for learning the similarity prediction function and randomly samples 30 classes ($\sim$39k images) from $ImageNet_{118}$ for the unlabeled target data. In this test, the backbone classification network is Resnet-18 and has weights initialized by classification on $ImageNet_{882}$. Both learning objectives (KCL and MCL) are optimized by SGD with mini-batch size 100.

**Results and Discussion:** We follow the evaluation procedure (including network architectures) used in Hsu et al. (2018), therefore the results can be directly compared. The results shown in Table 2 and 3 demonstrate a clear advantage for MCL over other methods. KCL also performs well, but MCL beats its performance with a larger gap when $C$ is unknown (ACC with K=100). MCL also estimates the number of classes in a dataset better than KCL (Appendix Table 5). The advantage of MCL over KCL in this section is not due to the ease of optimization, since the network is shallow in the Omniglot experiment and the network is pre-trained in the ImageNet experiment. The advantage may due to the fact that MCL is free of hyper-parameters and so performs better than KCL which uses a heuristic threshold ($\sigma = 2$) (Hsu & Kira, 2016) for its margin.

Table 3: Unsupervised cross-task transfer learning on ImageNet. The values (higher is better) are the average of three random subsets in $ImageNet_{118}$. Each subset has 30 classes. The "ACC" has $K = 30$. All methods use the features (outputs of average pooling) from Resnet-18 pre-trained with $ImageNet_{882}$ classification.

| Method | ACC | ACC(K=100) | NMI | NMI(K=100) |
|--------|-----|-----------|-----|-----------|
| K-means | 71.9% | 34.5% | 0.713 | 0.671 |
| LSC | 73.3% | 33.5% | 0.733 | 0.655 |
| LPNMF | 43.0% | 21.8% | 0.526 | 0.500 |
| KCL | 73.8% | 65.2% | 0.750 | 0.715 |
| MCL | **74.4%** | **71.5%** | **0.762** | **0.765** |

Table 4: Test error rates (lower is better) obtained by various semi-supervised learning approaches on CIFAR-10 with all but 4,000 labels removed. Supervised refers to using only 4,000 labeled samples from CIFAR-10 without any unlabeled data. All the methods use ResNet-18 and standard data augmentation.

| Method | CIFAR10 4k labels |
|--------|-------------------|
| Supervised | $25.4 \pm 1.0\%$ |
| Pseudo-Label | $19.8 \pm 0.7\%$ |
| Π-model | $19.6 \pm 0.4\%$ |
| VAT | $18.2 \pm 0.4\%$ |
| SPN-MCL | $22.8 \pm 0.5\%$ |
| Pseudo-MCL | $\mathbf{18.0} \pm 0.4\%$ |

## 5.4 SEMI-SUPERVISED LEARNING

We evaluate the semi-supervised learning performance of the Pseudo-MCL on the standard benchmark dataset CIFAR-10. The Pseudo-MCL is compared to two state-of-the-art methods, which are VAT (Miyato et al., 2018) and Π-Model (Laine & Aila, 2017; Sajjadi et al., 2016). Our list of baselines additionally includes Pseudo-Labeling (Lee, 2013) and SPN-MCL since they share a similar strategy with Pseudo-MCL. The SPN-MCL uses the same strategy presented in the Section 4.2 for unsupervised learning, except that the SPN is trained with only the labeled portion (*e.g.* 4k labeled data) of CIFAR10 in this section. We also note that the SPN serves as a static function to provide the similarity for optimizing the regular MCL objective.

**Experiment Setting:** To construct the $D_L$, four thousand labeled data are randomly sampled from the training set (50k images) of CIFAR10. This leaves 46k unlabeled data for $D_{UL}$. We use 5 random $D_L/D_{UL}$ splits to calculate the average performance. The images are augmented by the standard procedure which includes random cropping, random horizontal flipping, and normalization to zero mean with unit variance. The model for all method is the ResNet-18 (pre-activation version, He et al. (2016b)), which has no dropout as in a standard model. We use Adam to optimize the objective functions of all methods. The procedure begins with learning the supervised model with only the 4k labeled data; then all other methods have a fine-tuning with $D_L+D_{UL}$ based on the learned supervised model. The supervised model (with only 4k data) is trained with initial learning rate 0.001 and a decay with factor 0.1 at epochs 80 and 120 for a total of 140 epochs. All the semi-supervised methods are trained with initial learning rate 0.001 and have a decay with factor 0.1 at epoch 150 and 250 for a total of 300 epochs. We use a shared implementation among all methods so that the major difference between methods is the regularization term in the learning objective. Appendix C.1 provides the description for hyperparameter tuning.

**Results and Discussion:**

Table 4 presents the comparison and shows that Pseudo-MCL is on-par with the state-of-the-art method VAT (Miyato et al., 2018). The performance difference between SPN-MCL and Pseudo-MCL clearly demonstrates the benefits of having the binary classifier and the multi-class classifier optimized together. Note that comparing our Table 4 and a recent review (Oliver et al., 2018), we have a lower

baseline performance due to a lighter regularization (no dropout) and no extra data augmentation (such as adding Gaussian noise), but the relative ranking between methods is consistent. Therefore we confirm the effectiveness of Pseudo-MCL. Lastly, Pseudo-MCL is free of hyperparameter, which is a very appealing characteristic for learning with few data.

# 6 CONCLUSION

We presented a new strategy to learn a multi-class classification via a binary decision problem. We formulate the problem setting via a probabilistic graphical model and derive a simple likelihood objective that can be effectively optimized via neural networks. We show how this same framework can be used for three learning paradigms: supervised learning, unsupervised cross-task transfer learning, and semi-supervised learning. Results show comparable or improved results over state of the art, especially in the challenging unsupervised cross-task setting. This demonstrates the power of using pairwise similarity as weak labels to relax the requirement of class-specific labeling. We hope the presented perspective of meta classification inspires additional approaches to learning with fewer labeled data (e.g. domain adaptation and few-shot learning) as well as application to domains where weak labels are easier to obtain.

ACKNOWLEDGMENTS

This work was supported by the National Science Foundation and National Robotics Initiative (grant # IIS-1426998) and DARPA's Lifelong Learning Machines (L2M) program, under Cooperative Agreement HR0011-18-2-001.

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

## APPENDICES

## A  LOSS LANDSCAPE VISUALIZATION

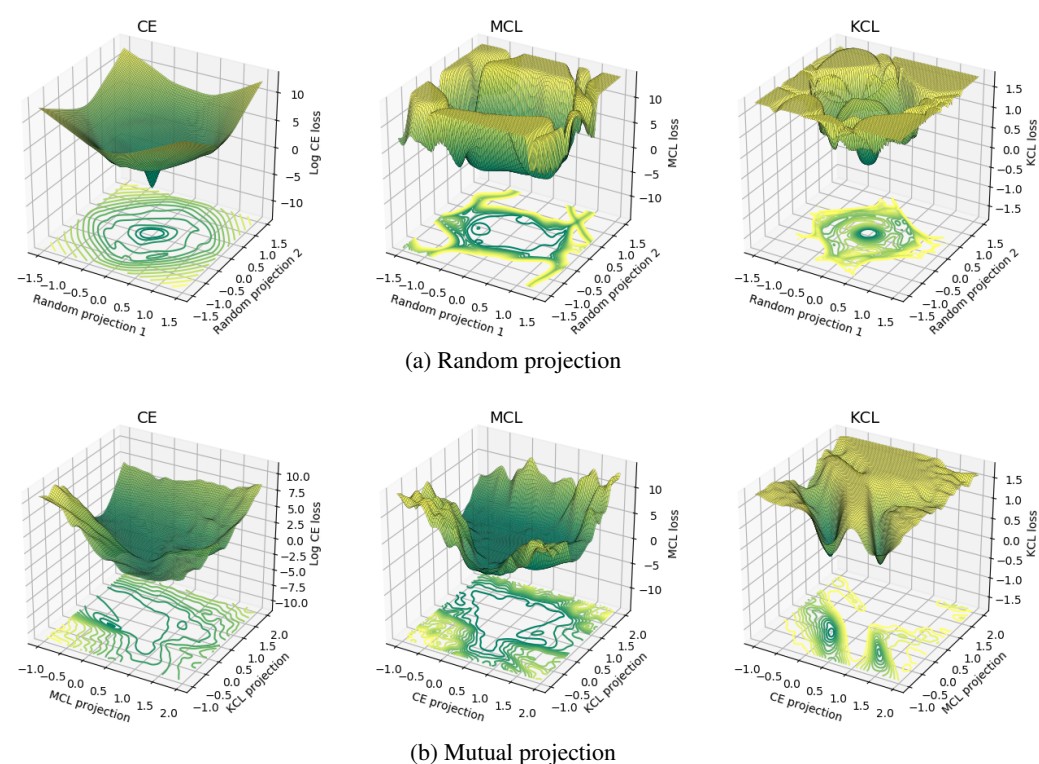

(a) Random projection

(b) Mutual projection

Figure 4: The loss landscape visualizations. Dark green represents a low loss value while yellow means high value. The bottom part of each diagram is the 2D contour of its 3D surface. The vertical axis of CE is logarithmic to better visualize its dynamic range (Li et al., 2017).

We visualize the three loss functions: CE, MCL, and KCL. The loss surfaces are plotted with the function (Goodfellow et al., 2014; Im et al., 2016; Li et al., 2017):

$$f(\alpha, \beta) = L(\theta^* + \alpha\delta + \beta\eta; D) \tag{8}$$

where $\theta^*$ are the parameters of the model trained with loss function $L$ and labeled dataset $D = (X, Y)$. The $\delta$ and $\eta$ variables are two directions for a 2D projection of $\theta$. The $\alpha$ and $\beta$ are the amount of shift along $\delta$ and $\eta$ from the origin $\theta^*$. This method allows us to better understand the landscape of loss around the solution.

To choose $\delta$ and $\eta$, one straightforward method is to use random projections. However, it cannot be used to compare the geometry across different networks or loss functions, because of the scale invariance in network weights. One source of such invariance is batch normalization. In such cases, the size (i.e., norm) of a filter (assume a convolution layer) is irrelevant because the output of each layer is re-scaled during batch normalization. Li et al. (2017) propose Filter-wise Normalization to address the above concern. We adopt this strategy to normalize the two random projections and make the relative flatness between loss surfaces comparable. We call this a random projection method.

Another way to choose $\delta$ and $\eta$ is to use solutions from different loss functions. Since we have three loss functions all able to solve the same multi-class classification problem, we can use one solution (e.g. $\theta^*_{MCL}$ from MCL) for the $\theta^*$ and use the remaining two solutions (e.g. $\theta^*_{CE}$ and $\theta^*_{KCL}$) for the two projections (e.g. $\delta = \theta^*_{CE} - \theta^*_{MCL}$ and $\eta = \theta^*_{KCL} - \theta^*_{MCL}$). We call this a mutual projection method.

**Visualization Setting:** This section uses CIFAR10 and VGG11. We choose VGG11 because it is the smallest network that KCL cannot be optimized well with a regular learning schedule. For each learning objectives, we use the best-learned models in that error rates are less than 10.4% (see Table 1). The parameters of three models ($\theta^*_{CE}$, $\theta^*_{MCL}$, $\theta^*_{KCL}$) are used to construct an interpolated one: $\theta = \theta^* + \alpha\delta + \beta\eta$. A 91x91 grid is used to enumerate the combinations of $\alpha$ and $\beta$, which are the scales for the two projected directions. The loss values associated with each ($\alpha$, $\beta$) are plotted in the z-direction to form a surface for visualization. Similar to Li et al. (2017), the vertical axis of CE is logarithmic to better visualize its dynamic range. For more details please refer to Li et al. (2017).

**Results and Discussion:** In the random projection (Figure 4a), the loss landscape with CE is similar to previous work (Li et al., 2017) which shows a nice convexity with a not-too-deep neural network (ResNet18). The solutions of MCL and KCL are both surrounded by a plateau of high loss, but MCL has a wider concave region. The same wide concavity can be seen in the mutual projection (Figure 4b). This is a possible explanation for why MCL still converges to a good local minimum with a randomly initialized network. Besides, the mutual projection shows that the geometry of MCL's loss landscape is similar to CE's surface, while KCL has a sharp low-loss region only around its solutions. This might be a reason why it requires a prolonged training schedule to find a good local minimum. Overall, MCL is qualitatively more similar to CE in the visualization of loss landscape.

## B   KCL VERSUS MCL

From the view of optimization objective, the KLD-based Contrastive Loss (KCL) has a form close to our MCL although it is originally designed for clustering. In the KCL paper (Hsu & Kira, 2016; Hsu et al., 2018), it interprets the softmax output of a neural network as outputting a probability distribution over cluster assignments. Then a contrastive loss function is defined using KL-divergence to measure the distance between two distributions $\hat{\mathbf{y}}_i = f(x_i; \theta)$ and $\hat{\mathbf{y}}_j = f(x_j; \theta)$. The cost between a similar pair $(x_i, x_j)$, in which $s_{ij} = 1$, is given by:

$$L^+_{KCL}(x_i, x_j) = D_{\mathrm{KL}}(\hat{\mathbf{y}}_i || \hat{\mathbf{y}}_j) + D_{\mathrm{KL}}(\hat{\mathbf{y}}_j || \hat{\mathbf{y}}_i). \tag{9}$$

If $(x_i, x_j)$ is a dissimilar pair ($s_{ij} = 0$), then $\hat{\mathbf{y}}_i$ and $\hat{\mathbf{y}}_j$ are expected to be different distributions, which is described by a hinge-loss function with a hyper-parameter $\sigma$ for the margin.

$$L^-_{KCL}(x_i, x_j) = L_h(D_{\mathrm{KL}}(\hat{\mathbf{y}}_i || \hat{\mathbf{y}}_j), \sigma) + L_h(D_{\mathrm{KL}}(\hat{\mathbf{y}}_j || \hat{\mathbf{y}}_i), \sigma),$$
$$\text{where } L_h(e, \sigma) = max(0, \sigma - e). \tag{10}$$

Then the total contrastive loss (KCL) has the form:

$$L_{KCL} = \sum_{i,j} s_{ij} L^+_{KCL}(x_i, x_j) + (1 - s_{ij}) L^-_{KCL}(x_i, x_j). \tag{11}$$

In comparing KCL and MCL, we find that they are similar in using pairwise similarity and have no requirement on the number of output nodes $K$ no matter what the true number of classes $C$ is. They can also be plugged into the training of neural networks in the same way, in that switching MCL to KCL can easily be done by replacing the learning criterion. Although they are similar in terms of usage, their formulation has a fundamental difference. KCL is inspired by metric learning, in that KL-divergence is the metric for evaluating the pairwise distance. Our MCL is inspired by the concept of meta classification learning and explained by a maximum likelihood estimation. The most significant difference is that MCL is free of hyperparameter. Therefore MCL does not require cross-validation for hyperparameter tuning. This property is crucial for unsupervised learning or when only a few instances of labeled data are available.

Table 5: Estimates for the number of characters across the 20 datasets in $Omniglot_{eval}$ when $C$ is unknown. The bold number means the prediction has error smaller or equal to 3. The number of dominant clusters is defined by $NDC = \sum_{i=1}^{K} [C_i >= E[C_i]]$, where $[\cdot]$ is an Iverson Bracket and $C_i$ is the size of cluster $i$. For example, $E[C_i]$ will be 10 if the alphabet has 1000 images and $K = 100$. The $ADif$ represents average difference (Hsu et al., 2018).

| Alphabet | #class | SKMS | KCL | MCL |
|---|---|---|---|---|
| Angelic | 20 | 16 | 26 | **22** |
| Atemayar Q. | 26 | 17 | 34 | **26** |
| Atlantean | 26 | 21 | 41 | **25** |
| Aurek_Besh | 26 | 14 | **28** | 22 |
| Avesta | 26 | 8 | 32 | **23** |
| Ge_ez | 26 | 18 | 32 | **25** |
| Glagolitic | 45 | 18 | **45** | 36 |
| Gurmukhi | 45 | 12 | **43** | 31 |
| Kannada | 41 | 19 | **44** | 30 |
| Keble | 26 | 16 | **28** | **23** |
| Malayalam | 47 | 12 | **47** | 35 |
| Manipuri | 40 | 17 | **41** | 33 |
| Mongolian | 30 | **28** | 36 | **29** |
| Old Church S. | 45 | 23 | **45** | 38 |
| Oriya | 46 | 22 | **49** | 32 |
| Sylheti | 28 | 11 | 50 | **30** |
| Syriac_Serto | 23 | 19 | 38 | **24** |
| Tengwar | 25 | 12 | 41 | **26** |
| Tibetan | 42 | 15 | **42** | 34 |
| ULOG | 26 | 15 | 40 | **27** |
| $ADif$ | | 16.3 | 6.35 | **5.1** |

## C EXPERIMENTAL SETTING

### C.1 HYPERPARAMETER TUNING FOR SEMI-SUPERVISED LEARNING

All the semi-supervised learning objectives $L_{SSL}$ here can be represented as a weighted sum of a supervised term $L_{sup}$ and an unsupervised regularization term $L_{reg}$:

$$L_{SSL} = \alpha L_{sup}(X_L, Y_L) + \beta L_{reg}(X_L \cup X_{UL}) \qquad (12)$$

For a fair comparison, one should give the same budget for tuning the hyperparameters, such as $\alpha$ and $\beta$. One strategy is applying an exhaustive grid search in the hyperparameter space. Such searching requires doing cross-validation and may not be applicable when the number of labeled data is small. We adopt another strategy that gives zero tuning budget for all. We decide the $\alpha$ and $\beta$ by natural statistics, which is the ratio between the amount of data be seen by the $L_{sup}$ and $L_{reg}$. Specifically:

$$\alpha = \frac{|D_L|}{|D| + |D_L|}, \quad \beta = \frac{|D|}{|D| + |D_L|} \qquad (13)$$

One method, VAT (Miyato et al., 2018), has extra hyperparameters (*e.g.* the $\epsilon$) in its design. In that case, we use the values decided in the original paper for this dataset.

## D ASSUMPTIONS IN META CLASSIFICATION LIKELIHOOD

### D.1 SIMPLIFIED LIKELIHOOD

In section 3, the original likelihood (eq. 2) relies on an additional independence assumption to simplify its negative logarithm form to a binary cross-entropy. Such an simplification raises the

question of whether equation (3) is over-simplified. For the supervised learning case (Section 4.1 with results in Section 5.2), where the constraints are ground truth, the global solution of our likelihood is also the solution for the original likelihood. This is because if an instance is misclassified, then it will break some pair-wise constraints in both likelihoods and no longer be optimal.

Of course, in practice, there could be two issues. First, the optimization methods for more complex models (e.g. stochastic gradient descent) may find local minima. Although it is hard to show theory for this in the general case, where local optima may be found, in such cases our visualization of the loss landscape (see Appendix A) provides some evidence that our method has a landscape that reduces poor local minima compared to prior work (KCL, Hsu et al. (2018)). The second potential issue is when constraints may be noisy. In such cases, for example, if the noise is high and there is a dependency structure to be leveraged, jointly optimizing across many or all constraints with the original likelihood may provide additional performance (at the expense of tractability). In practice, noisy constraints actually occur in our cross-task transfer learning experiments where our similarity prediction has significant errors (e.g. in Table 3 ImageNet experiments the similar pair precision, similar pair recall, dissimilar pair precision, and dissimilar pair recall are 0.812, 0.655, 0.982, and 0.992 respectively). The strong performance in terms of classification accuracy for the cross-task transfer experiments (Tables 2 and 3) shows that our simplification is robust to noise.

Overall, the fact that we have demonstrated our method on five image datasets and three application scenarios (Section 5.2 for supervised learning, 5.3 for unsupervised cross-task transfer learning, and 5.4 for semi-supervised learning) empirically support that the proposed likelihood can overcome these two issues. It would be interesting future work to develop methods that can incorporate constraints jointly, however.

## D.2    SEPARABILITY ASSUMPTIONS

Note that we assume separability of semantic categories in a dataset. This means that when the constraints are given (supervised learning), there is sufficient information (in the features) to separate or to group the samples. In the case of no given constraints (unsupervised or semi-supervised learning), there is also sufficient information to estimate the pairwise similarity. However, these are common assumptions that are inherent in discriminative models.

