# OpenReview forum: "Multi-class classification without multi-class labels"
_ICLR.cc/2019/Conference_

### Official Review · AnonReviewer1 · 2018-10-22
**Lacks citation to similar works in Statistics, topic modelling ...**

**Rating:** 5
**Confidence:** 4

**Review:**

The work is a special case of density estimation problems in Statistics, with a use of conditional independence assumptions to learn the joint distribution of nodes. While the work appears to be impressive, such ideas have typically been used in Statistics and machine learning very widely over the years(Belief Propagation,  Topic modeling with anchor words assumptions etc...). This work could be easily extended to multi-class classifications where each node belongs to multiple classes. It would be interesting to know the authors' thoughts on that. The hard classification rule in the paper seems to be too restrictive to be of use in practical scenarios, and soft classification would be a useful pragmatic alternative.

---

> ### Author Response · Authors · 2018-11-21
> **Discussion of potential extensions**
>
> We thanks the reviewer for raising the discussions of potential extensions and the nice comment of saying this work is impressive.
>
> [Q1]: "The work is a special case of density estimation problems in Statistics, with a use of conditional independence assumptions to learn the joint distribution of nodes. While the work appears to be impressive, such ideas have typically been used in Statistics and machine learning very widely over the years(Belief Propagation, Topic modeling with anchor words assumptions etc...)."
>
> We appreciate reviewers feedbacks from a high-level view of statistics and machine learning. Our main novelty, the reversed classifier encapsulation scheme, is derived from several concepts in the field. However, the way we combine these concepts is new, which includes a probabilistic graphical model, a non-linear discriminative model (deep neural networks), and parameter learning through a simplified likelihood. The effort of combining these ideas is for relaxing the requirement of the supervision in training a discriminative model. This is a different goal than topic modeling, which is often posed as an unsupervised learning problem.
>
> [Q2]: "The hard classification rule in the paper seems to be too restrictive to be of use in practical scenarios, and soft classification would be a useful pragmatic alternative. "
>
> Regarding the concern of the hard classification setting, we agree that in a certain application, such as document classification or community detection, that assigning a document or a person to only a single class/community could be restrictive for that purpose. However, there is also a large number of applications where this is not restrictive, e.g. in computer vision applications classifying an image or a pixel to only a class is one of the most common scenarios. Thus the concern of restrictiveness is application-dependent.
>
> [Q3]: "This work could be easily extended to multi-class classifications where each node belongs to multiple classes. It would be interesting to know the authors' thoughts on that."
>
> While our work focuses only on the problem of vanilla multi-class classification (hard classification, each sample can only belong to one class), it is interesting to discuss the possibility of extending it to a multi-label problem, where each sample can belong to multiple classes (labels). This would be interesting future work, but is not trivial because our problem reduction strategy requires a model to output a categorical distribution (has a sum of 1) so that the inner product of two categorical distributions (given two samples) represents the probability of being in the same class. In contrast, allowing a sample to belong to multiple classes means the sum of the model’s outputs can be as large as C (the number of classes) instead of 1; therefore the multi-label setting is not compatible with our problem reduction strategy as is and would require non-trivial modification.

---

### Official Review · AnonReviewer2 · 2018-10-31
**Very good paper**

**Rating:** 7
**Confidence:** 4

**Review:**

This paper proposed how to learn multi-class classifiers without multi-class labels. The main idea is shown in Figure 2, to regard the multi-class labels as hidden variables and optimize the likelihood of the input variables and the binary similarity labels. The difference from existing approaches is also illustrated in Figure 1, namely existing methods have binary classifiers inside multi-class classifiers while the proposed method has multi-class classifiers inside binary classifiers. The application of this technique to three general problem settings is discussed, see Figure 3.

Clarity: Overall, it is very well written. I just have two concerns.

First, the authors didn't discuss the underlying assumption of the proposed method except the additional independence assumption. I think there should be more underlying assumptions. For example, by the definition P(S_{i,j}=0 or 1|Y_i,Y_j) and the optimization of L(theta;X,S), does the "cluster assumption" play a role in it? The cluster assumption is popular in unsupervised/semi-supervised learning and metric learning where the X part of training data is in a form of pairs or triples. However, there is no such an assumption in the original supervised multi-class learning. Without figuring out the underlying assumptions, it is difficult to get why the proposed method works and when it may fail.

Second, there are too many abbreviations without full names, and some of them seem rather important such as KLD and KCL. I think full names of them should be given for the first time they appear. This good habit can make your audience more broad in the long run.

Novelty: As far as I know, the proposed approach is novel. It is clear that Section 3 is original. However, due to the writing style, it is hard to analyze which part in Section 4 is novel and which part is already known. This should be carefully revised in the final version. Moreover, there was a paper in ICML 2018 entitled "classification from pairwise similarity and unlabeled data", in which binary classifiers can be trained strictly following ERM without introducing the cluster assumption. The same technique can be used for learning from pairwise dissimilarity and unlabeled data as well as from pairwise similarity and dissimilarity data. I think this paper should be included in Section 2, the related work.

Significance: I didn't carefully check all experimental details but the experimental results look quite nice and promising. Given the fact that the technique used in this paper can be applied to many different tasks in machine learning ranging from supervised learning to unsupervised learning, I think this paper should be considered significant.

Nevertheless, I have a major concern as follows. In order to derive Eq. (2), the authors imposed an additional independence assumption: given X_i and X_j, S_{i,j} is independent of all other S_{i',j'}. Hence, Eqs. (2) and (3) approximately hold instead of exactly hold. Some comments should be given on how realistic this assumption is, or equivalently, how close (1) and (3) are. One more minor concern: why P(X) appears in (1) and then disappears in (2) and (3) when Y is marginalized?

---

> ### Author Response · Authors · 2018-11-21
> **Discussion of the assumptions**
>
> Thank you for the nice comment. We appreciate your effort in helping improve this work.
>
> [Q1]: "First, the authors didn't discuss the underlying assumption of the proposed method except the additional independence assumption. I think there should be more underlying assumptions. … does the "cluster assumption" play a role in it?"
>
> While we agree that we do have some underlying assumptions, it is different than the clustering one. For example, we assume separability of semantic categories. This means that when the constraints are given (supervised learning), there is sufficient information (in the features) to separate or to group the samples. In the case of no given constraints (unsupervised or semi-supervised learning), there is also sufficient information to estimate the pairwise similarity. However, these are common assumptions that are inherent in discriminative models. We added Appendix D.2 to include the above discussions.
>
> [Q2]: “Second, there are too many abbreviations without full names, and some of them seem rather important such as KLD and KCL.” and “due to the writing style, it is hard to analyze which part in Section 4 is novel and which part is already known.”
>
> Thanks for pointing out the writing issues such as the use of abbreviations and the claims of novelty. We fixed the abbreviations and added extra lines at the beginning of Section 4 and related work to clarify the novel parts. We also made an update to add the mentioned ICML paper into the related works.
>
> [Q3]: In order to derive Eq. (2), the authors imposed an additional independence assumption: given X_i and X_j, S_{i,j} is independent of all other S_{i',j'}. Hence, Eqs. (2) and (3) approximately hold instead of exactly hold. Some comments should be given on how realistic this assumption is, or equivalently, how close (1) and (3) are.
>
> For the supervised learning case (Section 4.1 with results in Section 5.2), where dense ground truth constraints are available, the global solution of our likelihood is also the solution for the original likelihood. This is because if an instance is misclassified, then it will break some pair-wise constraints in both likelihoods and no longer be optimal.
>
> Of course, in practice, there could be two issues. First, the optimization methods for more complex models (e.g. stochastic gradient descent) may find local minima. Although it is hard to show theory for this in the general case, where local optima may be found, in such cases our visualization of the loss landscape (see Appendix A) provides some evidence that our method has a landscape that reduces poor local minima compared to prior work (KCL (Hsu et al., 2018)). The second potential issue is when constraints may be noisy. In such cases, for example, if the noise is high and there is a dependency structure to be leveraged, jointly optimizing across many or all constraints with the original likelihood may provide additional performance (at the expense of tractability). In practice, noisy constraints actually occur in our cross-task transfer learning experiments where our similarity prediction has significant errors (e.g. in Table 3 ImageNet experiments the similar pair precision, similar pair recall, dissimilar pair precision, and dissimilar pair recall are 0.812, 0.655, 0.982, and 0.992 respectively). The strong performance in terms of classification accuracy for the cross-task transfer experiments (Tables 2 and 3) shows that our simplification is robust to noise.
>
> Overall, the fact that we have demonstrated our method on five image datasets and three application scenarios (Section 5.2 for supervised learning, 5.3 for unsupervised cross-task transfer learning, and 5.4 for semi-supervised learning) empirically support that the proposed likelihood can overcome these two issues. It would be interesting future work to develop methods that can incorporate constraints jointly, however. We added Appendix D.1 to include the above discussions.
>
> [Q4]: "One more minor concern: why P(X) appears in (1) and then disappears in (2) and (3) when Y is marginalized?"
>
> The reason we omit the P(X) in equation (2) and (3) is because the Xs are observed leaf nodes which do not affect the optimization of the likelihood. We have clarified this in the revision.

---

> > ### Comment · AnonReviewer2 · 2018-11-30
> > **The authors have clarified basically all my concerns**
> >
> > Only a minor question: isn't "separability of semantic categories" the cluster assumption in semi-supervised learning, also known as the global consistency, one of the three famous consistencies in semi-supervised learning (the other two are local consistency and perturbation consistency)?

---

> > > ### Author Response · Authors · 2018-12-05
> > > **Good question**
> > >
> > > Thanks for raising the nice discussion; yes the cluster assumption is related. In our opinion, the cluster assumption implies separability but additionally assumes that the data distribution has a higher density in a semantic category and a lower density between categories. Since our method is driven by the constraints (either given or learned), this does not necessarily have to be the case as long as there is enough information in the features to separate the categories. It could be an interesting future work to include such additional assumptions, for example by adopting a large margin criterion.

---

### Official Review · AnonReviewer3 · 2018-11-03
**The paper introduces some novel ideas but lacks elaborate justification.**

**Rating:** 6
**Confidence:** 3

**Review:**

In this paper the authors revisit the problem of multi-class classification and propose to use pairwise similarities (more accurately, what they use is the co-occurrence pattern of labels) instead of node labels. Thus, having less stringent requirements for supervision, their framework has broader applicability: in supervised and semi-supervised classification and in unsupervised cross-task transfer learning, among others.

Pros: The idea of using pairwise similarities to enable a binary classifier encapsulate a multi-class classifier is neat.

Cons: My main gripe is with the conditional independence assumption on pairwise similarities, which the author use to simplify the likelihood down to a cross-entropy. Such an assumption seems too simple to be useful in problems with complicated dependence structure. Yes, the authors conduct some experiments to show that their algorithms achieve good performance in some benchmark datasets, but a careful discussion (if possible, theoretical) of when such an assumption is viable and when it is an oversimplification is necessary (analogous assumptions are used in naive Bayes or variational Bayes for simplifying the likelihood, but those are much more flexible, and we know when they are useful and when not).

Secondly, by using co-occurrence patterns, one throws away identifiability---the (latent) labels are only learnable up to a permutation unless external information is available. This point is not made clear in the paper, and the authors should describe how they overcome this in their supervised classification experiments.

---

> ### Author Response · Authors · 2018-11-21
> **Elaboration**
>
> Thank you for your insightful comments. We appreciate your acknowledgment of the novelty, and we are glad to elaborate on the two concerns.
>
> [Q1]:  "Such an assumption seems too simple to be useful in problems with complicated dependence structure … a careful discussion (if possible, theoretical) of when such an assumption is viable and when it is an oversimplification is necessary"
>
> For the supervised learning case (Section 4.1 with results in Section 5.2), where dense ground truth constraints are available, the global solution of our likelihood is also the solution for the original likelihood. This is because if an instance is misclassified, then it will break some pair-wise constraints in both likelihoods and no longer be optimal.
>
> Of course, in practice, there could be two issues. First, the optimization methods for more complex models (e.g. stochastic gradient descent) may find local minima. Although it is hard to show theory for this in the general case, where local optima may be found, in such cases our visualization of the loss landscape (see Appendix A) provides some evidence that our method has a landscape that reduces poor local minima compared to prior work (KCL (Hsu et al., 2018)). The second potential issue is when constraints may be noisy. In such cases, for example, if the noise is high and there is a dependency structure to be leveraged, jointly optimizing across many or all constraints with the original likelihood may provide additional performance (at the expense of tractability). In practice, noisy constraints actually occur in our cross-task transfer learning experiments where our similarity prediction has significant errors (e.g. in Table 3 ImageNet experiments the similar pair precision, similar pair recall, dissimilar pair precision, and dissimilar pair recall are 0.812, 0.655, 0.982, and 0.992 respectively). The strong performance in terms of classification accuracy for the cross-task transfer experiments (Tables 2 and 3) shows that our simplification is robust to noise.
>
> Overall, the fact that we have demonstrated our method on five image datasets and three application scenarios (Section 5.2 for supervised learning, 5.3 for unsupervised cross-task transfer learning, and 5.4 for semi-supervised learning) empirically support that the proposed likelihood can overcome these two issues. It would be interesting future work to develop methods that can incorporate constraints jointly, however.
>
> We added Appendix D.1 to include the above discussions.
>
> [Q2]: "Secondly, by using co-occurrence patterns, one throws away identifiability---the (latent) labels are only learnable up to a permutation unless external information is available. This point is not made clear in the paper ..."
>
> To address the concern about the identifiability of clusters, we slightly augment the second paragraph of Section 5.1 with additional references. In summary, for the supervised classification experiments, we use the Hungarian assignment algorithm to assign clusters to labels given the ground truth information (this is commonly used to evaluate clustering algorithms, e.g. see (Yang et al., 2010)). When labels are not available (e.g. in cross-task transfer learning) we only do this type of assignment for quantitative evaluation purposes.
>
> We again thank the reviewer’s effort for improving this work.

---

### Meta-Review · Area_Chair1 · 2018-12-17
**Interesting contribution to multi-class learning**

**Confidence:** 4
**Recommendation:** Accept (Poster)

**Metareview:**

This paper provides a technique to learn multi-class classifiers without multi-class labels, by modeling the multi-class labels as hidden variables and optimizing the likelihood of the input variables and the binary similarity labels.

The majority of reviewers voted to accept.